# Effect of Betaine Diet on Growth Performance, Carcass Quality and Fat Deposition in Finishing Ningxiang Pigs

**DOI:** 10.3390/ani11123408

**Published:** 2021-11-29

**Authors:** Yaodong Wang, Jiayi Chen, Yingli Ji, Xue Lin, Yurong Zhao

**Affiliations:** College of Animal Science & Technology, Hunan Agricultural University, Changsha 410125, China; sxllwyd@gmail.com (Y.W.); jiayi@stu.hunau.edu.cn (J.C.); jiyingli168@163.com (Y.J.); risca@tanke.com.cn (X.L.)

**Keywords:** betaine, carcass quality, fat deposition, Ningxiang pig, meat quality

## Abstract

**Simple Summary:**

Excessive fat deposit is one of the major problems in finishing Ningxiang pigs, and adversely affects the breeding. The study aimed to investigate the effects of diet with betaine supplementation (basal diet + 0.2% betaine) on the growth performance, slaughter performance, meat quality and the genes expression related to fat deposition in finishing Ningxiang pigs. The results indicated that diet with betaine supplementation decreased back fat thickness and fat percentage, and increased the lean meat percentage as well. In addition, it reduced the fat deposition by regulating the genes expression. These findings provided a reference for breeding Ningxiang pigs.

**Abstract:**

The present study was conducted to investigate the effects of diet with betaine supplementation on the growth performance, carcass quality and fat deposition in finishing Ningxiang pigs. A total of 24 Ningxiang pigs (43.6 ± 5.34 kg of average body weight) was randomly divided into two groups, with 6 replicates per treatment and 2 pigs per replicate. The treatments included a control group (basal diet) and a test group (basal diet + 0.2% betaine). The whole trial lasted 81 days. At the end of the experiment, one pig (close to the average body weight of all experimental pigs) per replicate was slaughtered to determine the carcass traits, meat quality and the mRNA expression levels of genes relate to fat deposition (one pig per replicate was randomly selected and fasted for 12 h, *n* = 6). Results indicated that growth performance was not changed with betaine supplementation. However, dietary with betaine supplementation decreased back fat thickness and fat percentage, and increased the lean meat percentage as well (*p* < 0.05). In addition, diet with betaine supplementation reduced drip loss, water loss, cooking loss, shear force and b × 24 h value of meat (*p* < 0.05). There was no difference in total moisture, ether extract and crude protein of longissimus thoracis between the control and test group. Dietary with betaine supplementation decreased ether extract and total cholesterol (*p* < 0.05) in liver. Dietary with betaine supplementation upregulated the mRNA expression levels of adipose triglyceride lipase (ATGL) and sirtuin 1 (Sirt1), while downregulated the mRNA expression levels of fatty acid synthase (FAS) and acetyl CoA carboxylase (ACC) in subcutaneous fat of back (*p* < 0.05). Besides, dietary with betaine supplementation upregulated the fatty acid binding protein 4 (FABP4) mRNA expression of longissimus thoracis in finishing Ningxiang pigs (*p* < 0.05). These results showed that diet supplemented with betaine could improve the slaughtering performance and meat quality, and regulate the genes expression to affect the fat deposition in finishing Ningxiang pigs.

## 1. Introduction

Fat is an important tissue for storing energy, and is closely related to livestock and poultry production and meat quality [1]. Excessive fat deposition will decrease the lean meat percentage, the meat quality and the ratio of feed conversion, and even cause serious health problems such as fatty liver [2].

Ningxiang pig, as a local pig breed of China, occupies an important position in Chinese pork consumption with its delicious taste—as a fat-type pig, it has strong ability to deposit fat [3]. Lei et al. [4] reports that Ningxiang pigs exhibit higher back fat thickness than Large White pigs (4.54% vs. 2.61%). He et al. [5] conclude that Ningxiang pigs have higher fat mass (42.3% vs. 21.9%) and lower muscle mass (35.4% vs. 58.9%) than Duroc × Landrace × Large Yorkshire pigs. Therefore, it is necessary to find a way to decrease the fat deposition in Ningxiang pigs for high feed conversion efficiency with lower cost. As an intermediate product of metabolism process, betaine generally exists in animals and plants. With the further maturity of betaine chemical synthesis method and the continuous reduction of production cost, betaine has been widely used in livestock and poultry production [6]. Fernández et al. [7] found that betaine supplementation increased lean deposition in growing Iberian gilts. Therefore, it was hypothesized that betaine supplementation may reduce fat deposition and improve lean meat percentage in finishing Ningxiang pigs. The study was conducted to explore the effect of betaine supplementation on growth performance, carcass quality and fat metabolism in finishing Ningxiang pigs, which provide theoretical basis for the development and utilization of betaine in Ningxiang pigs.

## 2. Materials and Methods

### 2.1. Diet and Animal Management

The purity of betaine (Skystone Feed Co., Ltd., Yixing, China) was 96%. This experiment was conducted in Bangshifu pig farm (Changsha, China) and the study was approved by the Biomedical Research Ethics Committee of Hunan Agricultural University (the authorization number from ethical committee is 2021, No. 91). A total of 24 Ningxiang pigs (43.6 ± 5.34 kg of average body weight) was randomly divided into two groups, with 6 replicates per treatment and 2 pigs per replicate (1:1 ratio of gilt and boar). The whole trial lasted 81 days. Pigs were fed three times per day with free access to water. The treatments included a control group (basal diet) and a test group (basal diet + 0.2% betaine). Basis diets were designed with the recommendations of Chinese National Feeding Standard Type 2 for fatty growing pigs (2020), as shown in Table 1.

### 2.2. Sampling and Measurements

Pigs were weighed at the beginning and the end of the experiment, individually. The feed intake was recorded weekly and calculated for each pen. The average daily gain (ADG), average daily feed intake (ADFI), and feed conversion ratio (FCR) were then calculated using this information for each phase.

At the end of the experiment, one pig (close to the average body weight of all experimental pigs) per replicate was slaughtered to determine the carcass traits and meat quality (one pig per replicate was randomly selected and fasted for 12 h, *n* = 6). Samples of longissimus thoracis (LT), subcutaneous fat of back and liver were collected from the right side of the carcass stored at liquid nitrogen, and then stored at −80 °C for further analysis.

The carcass traits included: slaughter rate (%) = (slaughter weight/live weight) × 100; carcass weight: the body weight after slaughter with removed head, hoof, tail and viscera and retained the oil and kidney; fat rate (%) = (fat weight/live weight) × 100; lean rate (%) = (lean weight/carcass weight) × 100; back fat thickness: take the average back fat thickness of shoulder, thoracolumbar junction and lumbar sacral junction of the left side of carcass.

Pork quality was determined according to the method of Honikel [9]. Meat color (lightness, redness and yellowness) was determined after slaughter (45 min and 24 h) by using reflectance spectrophotometer (CR-410, Konica Minolta Sensing Inc., Osaka, Japan). Cooking loss was determined by using weighted meat sample placed in a pot cooked to a core temperature of 70 °C, and then taken out and cooled for 30 min before weight. The ratio of the reduced weight of the meat sample to the initial weight was the cooking loss. The cooking loss, shearing force, drip loss and water loss were measured according to the method described by Chen [10].

Total moisture, ether extract (EE) of liver and longissimus thoracis samples were analyzed referring to the Association of Analytical Chemists methods (AOAC 2000) [11]. Triglyceride (TG) and total cholesterol (TC) was measured using TG and TC kit (Nanjing Jiancheng Bioengineering Institute) according to the manufacturer’s protocol.

### 2.3. Quantitative Real-Time PCR Analysis

Total RNA was isolated from longissimus thoracis and subcutaneous fat of back with Trizol reagent (Beijing Solarbio Science & Technology Co., Ltd., Beijing, China) according to the method described by Chen [10]. The purity and integrity of RNA were detected by Nanodrop (Thermo Scientific, Waltham, Massachusetts, USA) and electrophoresis in 1% agarose gel, respectively. Reverse transcription of the total RNA with RT Reagents (Hunan Accurate Biotechnology Co., Ltd., Changsha, China). The qPCR cycling program were as follows: an initial denaturation at 95 °C for 30 s and 40 cycles of denaturation at 95 °C for 5 s and annealing and extension at 60 °C for 30 s. The relative expression levels of mRNA were calculated by using the 2^−(^^ΔΔCT)^ method [12]. The considered genes include acetyl-CoA carboxylase (ACC), fatty acid synthetase (FAS), adipose triglyceride lipase (ATGL), fatty acid-binding protein4 (FABP4) and sirtuin1 (Sirt1). β-actin was chosen as the housekeeping gene. The primers were designed via Primer 5.0 software; sequences of primers are shown in Table 2.

### 2.4. Statistical Analysis

All data were analyzed to evaluate whether data were normally distributed by using IBM SPSS Statistics 22.0. The repeated measurements were tested by using ANOVA. Differences among means were tested by using the method of the T-test. Results were expressed as the mean ± standard error of means (SEM). *p* ≤ 0.05 was considered to be statistically significant.

## 3. Results

### 3.1. Growth Performance

There was no difference of ADG, ADFI and FCR between the control and test group (Table 3).

### 3.2. Slaughtering Performance

Diet with betaine supplementation decreased fat percentage (*p* < 0.05; Table 4), back fat thickness (*p* < 0.05) and had no effect on slaughter rate.

### 3.3. Meat Quality

Diet with betaine supplementation decreased shearing force, drip loss and cooking loss (*p* < 0.05; Table 5). It also decreased b*_24h_ value (*p* < 0.05) with no effect on redness and lightness.

### 3.4. Longissimus Thoracis Chemical Composition and Biochemical Parameters

There was no difference in total moisture, ether extract and crude protein of longissimus thoracis between the control and test group (Table 6).

### 3.5. Liver Chemical Composition and Biochemical Parameters

Dietary with betaine supplementation decreased ether extract and total cholesterol (*p* < 0.05; Table 7) in liver.

### 3.6. Relative mRNA Expression Levels

The relative mRNA expression levels of FABP4 upregulated in longissimus thoracis of the BET group (*p* < 0.05; Figure 1). There was no difference in the relative mRNA expression levels of FAS and ACC between two groups.

As shown in Figure 2, in subcutaneous back fat of the BET group, the relative mRNA expression levels of ACC and FAS downregulated significantly (*p* < 0.05) and the mRNA expression of ATGL and Sirt1 upregulated (*p* < 0.05). However, the expression of FABP4 showed no difference when compared to the control group.

## 4. Discussion

There were studies reporting that diet with betaine supplementation might upregulate the growth performance in pigs [13]. However, there were no differences of ADG, ADFI and FCR between the control and test group in this study, which were in line with Albuquerque et al. [14] and Sale et al. [15].

Meat quality is an important actor for livestock and poultry production and used to be evaluated by meat color, shearing force, drip loss, cooking loss and others. In general, meat color is the first consideration for people to make a purchase decision, although meat color is not the most reliable forecaster for its safety and quality [16]. Besides, the percentage of intramuscular fat (IMF) is also the important actor of meat quality. Additionally, there were researches reporting the positive effect of IMF on meat quality [17,18,19]. Meat color depends on the content of meat pigment, which is mainly composed of myoglobin, hemoglobin and trace-colored metabolites. Yu et al. [20] found that meat color scores were correlated to the proportion of oxymyoglobin, and negatively correlated with deoxymyoglobin and metmyoglobin content. Therefore, the meat color has a close relationship with the quality of meat. This study presented no marked change on a* and L* value of meat, which was consistent with the result of other research [21]. However, b*_24h_ value of the BET diet group was markedly decreased. Thus, it indicated that betaine diet has a positive effect on the meat color. Besides, the study found that diet with betaine supplementation decreased the drip loss, cooking loss and shear force of the LT muscle, while increasing the water-holding capacity and tenderness of the muscles. These were consistent with other studies that betaine can maintain the balance of water in the cell by the special structure, so that betaine diet can greatly affect water-holding capacity in meat [22,23]. As a result of the less exudated water distributing on the surface of the muscle fiber, the ability to reflect light was weakened, thus b*_24h_ value of meat with betaine diet decreased [24]. The present study indicated that the expression of FABP4 in LT muscles of the BET diet group was significantly upregulated. However, it was found that there was no difference in ether extract on longissimus thoracis between the control and test group. As a candidate gene for IMF deposition [25], FABP4 affects metabolic pathways by regulating transport of fatty acids and other ligands [26,27]. The present results indicated that diets supplemented with betaine upregulated the expression of FABP4. However, it had no negative effect on ether extract of longissimus thoracis. In summary, dietary with betaine supplementation could affect the meat quality via decreasing b*_24h_ value of meat, increasing the water-holding capacity and tenderness of the muscles. Additionally, diet with betaine supplementation has no negative effect on ether extract of muscle.

Present results indicated that diet with betaine supplementation decreased the levels of EE and TC in liver, which was consistent with previous studies [28]. Fat is an important part of the body, playing a key role in storing and supplying energy, maintaining body temperature, fixing internal organs and promoting the absorption of fat-soluble vitamins. However, excessive fat deposition makes negative influence on growth and even causes illness [29]. As the main metabolic organ of the body, the liver is an important center of fat synthesis and decomposition. Betaine is involved in the fat oxidation in the liver. In addition, betaine can supply methyl for the body to produce lecithin and carnitine, which can promote the metabolization of fat in liver [30,31]. Thus, dietary with betaine supplementation decreased the level of TC in the liver.

Fat metabolism includes the deposition, synthesis, decomposition and transport. Fat deposition is actually the proliferation and growth of adipocytes. The key to control the proliferation and growth of adipocytes is to regulate the synthesis and decomposition of fat. ACC is the rate-limiting enzyme for fatty acid de novo synthesis [32,33]. Previous study finds that betaine can decrease fat deposition and improve the lean meat percentage of animals [34]. Present results indicated that dietary with betaine supplementation decreased fat percentage and back fat thickness. The reason for the decrease of fat deposition might be related to genes expression. The present study indicated that dietary with betaine supplementation regulated the genes mRNA expression in subcutaneous back fat (downregulated the ACC and FAS mRNA expression and upregulated the ATGL and Sirt1 mRNA expression), and the results were in line with the findings previously described in literature [35]. Therefore, the mechanism might be related to the function of modification of DNA methylation with betaine. Besides, there are reports found that choline diet could mitigate excessive fat deposition, while betaine could change into choline in body [36,37]. Kim et al. [38] reported that the ACC mRNA expression in viral-induced mice downregulated after diet with betaine was supplemented. Another study indicated gene methylation levels of the ACC in diet with betaine supplementation group mice was higher than control group, which verified betaine may regulate the expression of ACC by supplying methyl [39]. Besides, diet with betaine supplementation can also regulate the expression of ACC by regulating other genes [30,31]. FAS, the finally rate-limiting enzyme for fatty acid de novo synthesis, mainly exist in the liver and fat tissues, which can catalyze the synthesis of acetyl-CoA and malonyl-CoA into long-chain fatty acids [40]. High-fat diet promotes high expression of FAS [41]. Current results had verified betaine dietary downregulated the FAS and ACC mRNA expression in fat by Yang et al. [42] and Dong et al. [43]. ATGL can regulate the deposition of fat by regulating the process of triglyceride decomposition [44]. It is reported that systemic and adipocyte-specific ATGL knockout mice exhibit moderate obesity [45]. In addition, ATGL can promote the expression of Sirt1 [46]. Sathyanarayan et al. [47] found the high expression of ATGL on mice showed excessive fat deposition, and the situation was found to alleviate after the knockdown of Sirt1, which proved that the function of ATGL on fat metabolism appears to be mediated by Sirt1. Zhao et al. [48] found that betaine diet could upregulate the ATGL mRNA expression in fat by modification of DNA methylation. In summary, the present study suggested that betaine could mitigate excessive fat deposition by downregulating the expression of ACC and FAS and upregulating the expression of ATGL and Sirt1.

## 5. Conclusions

In conclusion, diet supplemented with betaine could decrease fat deposition via regulating the expression of genes related to lipid metabolism. In addition, diet supplemented with betaine could improve meat quality via increasing water-holding capacity and tenderness in muscles of finishing Ningxiang pigs.

## Figures and Tables

**Figure 1 animals-11-03408-f001:**
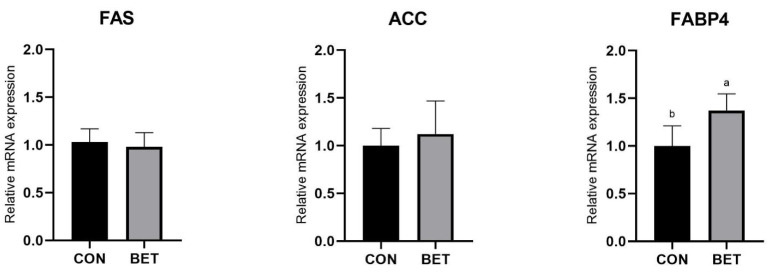
mRNA expression of longissimus thoracis (LT) in finishing Ningxiang pigs fed with betaine. The relative mRNA expression levels of the key genes related to fat metabolism including acetyl-CoA carboxylase (ACC), fatty acid synthetase (FAS), fatty acid-binding protein4 (FABP4) (*n* = 6). CON, control diet group. BET, betaine diet group. ^a, b^ mean in the same row with different letter superscripts signal a significant difference, ^a^ as the highest numerical value (*p* ≤ 0.05).

**Figure 2 animals-11-03408-f002:**
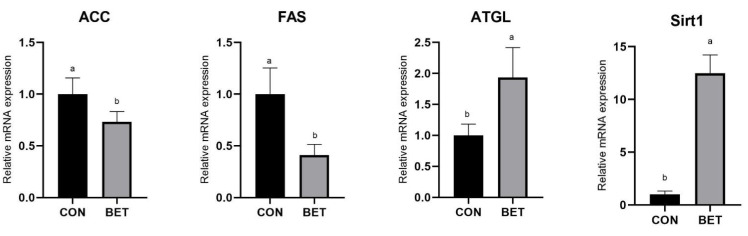
mRNA expression of fat metabolism related genes in subcutaneous back fat. The relative mRNA expression levels of the key genes related to fat metabolism including acetyl-CoA carboxylase (ACC), fatty acid synthetase (FAS), adipose triglyceride lipase (ATGL), fatty acid-binding protein4 (FABP4), sirtuin1 (Sirt1) (*n* = 6). CON, control diet group. BET, betaine diet group. ^a, b^ mean in the same row with different letter superscripts signal a significant difference, ^a^ as the highest numerical value (*p* ≤ 0.05).

**Table 1 animals-11-03408-t001:** Composition and nutrient levels of basal diets (air-dry basis) ^1^.

Ingredients (%)	CON	BET
Corn	54.90	54.90
Wheat bran	21.30	21.30
Soybean meal	8.20	8.20
Rice bran	6.40	6.40
Corn oil	3.20	3.20
Zeolite	1.90	1.69
Straw	1.70	1.70
Limestone	1.01	1.01
Premix ^2^	1.00	1.00
CaHPO_4_	0.27	0.27
Lysine, 98%	0.11	0.11
Threonine 98%	0.01	0.01
Betaine 96%	0.00	0.21
Total	100.00	100.00
Nutrient levels ^3^		
Digestible energy, MJ/kg	13.03	13.03
Crude protein, %	12.01	12.01
Crude fat, %	5.28	5.28
Crude Fiber, %	4.07	4.07
Lysine, %	0.60	0.60
Total phosphorus %	0.53	0.53
Calcium, %	0.50	0.50
Threonine, %	0.44	0.44
Methionine, %	0.20	0.20
Available phosphorus, %	0.16	0.16
Tryptophan, %	0.13	0.13

^1^ Basal diet formulated according to the Chinese National Feeding Standard for Swine. ^2^ The premix supplied, per kg of diet: VA 1300 IU, VD3 150 IU, VE 11IU, VK3 0.5 mg, VB1 1.2 mg, VB2 2 mg, VB6 1.3 mg, VB12 5 μg, folic acid 0.3 mg, pantothenic acid 7 mg, Cu 5 mg, I 3 mg, Se 0.15 mg, Zn 80 mg, Fe 80 mg, Mn 60 mg, NaCl 3600 mg. ^3^ The contents of digestible energy, crude protein, crude fiber, calcium and total phosphorus were analyzed, Nutrient levels were calculated according to the tables of feed composition and nutritive values in China (30th edition in 2019) [8]. CON, control diet group. BET, betaine diet group.

**Table 2 animals-11-03408-t002:** Primers used for quantitative real-time PCR.

Gene	Accession No.	Primer Sequence (5′–3′)	Product Size (bp)
β-ACTIN	NM_001170517.2	F:CCTGCGGCATCCACGAAAC	123
		R:TGTCGGCGATGCCTGGGTA	
ACC	NM-001114269	F:GGCCATCAAGGACTTCAACC	120
		R:ACGATGTAAGCGCCGAACTT	
FAS	NM_001099930.1	F:ACACCTTCGTGCTGGCCTAC	112
		R:ATGTCGGTGAACTGCTGCAC	
FABP4	NM_001002817.1	F:GGGCCAGGAATTTGATGAAG	103
		R:CTTTCCATCCCACTTCTGCAC	
ATGL	NM_001098605.1	F:CATCCGTGGCTGCCTGGTGAA	127
		R:CCTGGCGGCGAAGTGGGTTAT	
Sirt1	NM_001145750.2	F:GCAGGAATCCAGAGG	281
		R:GGTCTTACTTTCAGGGA	

ACC, acetyl-CoA carboxylase. FAS, fatty acid synthetase. ATGL, adipose triglyceride lipase. FABP4, fatty acid-binding protein4. Sirt1, sirtuin1.

**Table 3 animals-11-03408-t003:** Effect of betaine on growth performance of finishing Ningxiang pigs.

Items	CON	BET	SEM	*p* Value
Initial body weight (kg)	44.81	42.38	1.334	0.379
Final body weight (kg)	72.50	72.44	1.309	0.982
Average daily gain (ADG, g)	0.34	0.37	0.017	0.430
Average daily feed intake (ADFI, kg)	1.65	1.74	0.044	0.287
Feed conversion ratio (FCR)	4.89	4.79	0.142	0.746

CON, control diet group. BET, betaine diet group. ADG, average daily gain. ADFI, average daily feed intake. FCR, feed conversion ratio.

**Table 4 animals-11-03408-t004:** Effect of betaine on slaughtering performance of finishing Ningxiang pigs.

Items	CON	BET	SEM	*p* Value
Live weight (kg)	73.92	74.17	1.312	0.929
Carcass weight (kg)	54.72	54.55	1.231	0.949
Slaughter rate (%)	73.95	73.55	0.682	0.787
Fat percentage (%)	39.50 ^a^	36.40 ^b^	0.680	0.014
Lean meat percentage (%)	39.53 ^a^	42.23 ^b^	0.685	0.042
Back fat thickness (mm)	39.69 ^a^	34.92 ^b^	1.062	0.016

CON, control diet group. BET, betaine diet group. ^a, b^ mean in the same row with different letter superscripts signal a significant difference, ^a^ as the highest numerical value (*p* ≤ 0.05).

**Table 5 animals-11-03408-t005:** Effect of betaine on the meat quality of finishing Ningxiang pigs.

Items		CON	BET	SEM	*p* Value
Redness	a*_45 min_	10.41	10.11	0.538	0.800
	a*_24 h_	11.33	10.63	0.606	0.588
Yellowness	b*_45 min_	6.09	4.99	0.389	0.166
	b*_24 h_	6.30 ^a^	5.06 ^b^	0.290	0.025
Lightness	L*_45 min_	46.27	42.75	2.119	0.432
	L*_24 h_	47.08	43.92	1.801	0.407
Shearing force (N)		43.42 ^a^	39.65 ^b^	0.699	0.001
Drip loss (%)		1.80 ^a^	1.16 ^b^	0.134	0.009
Cooking loss (%)		35.77 ^a^	32.77 ^b^	0.693	0.021

CON, control diet group. BET, betaine diet group. ^a, b^ mean in the same row with different letter superscripts signal a significant difference, ^a^ as the highest numerical value (*p* ≤ 0.05).

**Table 6 animals-11-03408-t006:** Effect of betaine on the longissimus thoracis chemical composition and biochemical parameters of finishing Ningxiang pigs.

Items	CON	BET	SEM	*p* Value
Total moisture (%)	74.78	75.52	0.592	0.567
Ether extract (EE, %)	2.12	2.08	0.064	0.818
Crude protein (CP, %)	22.48	21.92	0.369	0.475

CON, control diet group. BET, betaine diet group. EE, ether extract. CP, crude protein.

**Table 7 animals-11-03408-t007:** Effect of betaine on the liver chemical composition and biochemical parameters of finishing Ningxiang pigs.

Items	CON	BET	SEM	*p* Value
Total moisture (%)	71.28	70.93	0.098	0.068
Ether extract (EE, %)	3.80 ^a^	2.60 ^b^	0.224	0.002
Triglyceride (TG, mmol/L)	2.82	2.88	0.094	0.751
Total cholesterol (TC, mmol/L)	0.98 ^a^	0.75 ^b^	0.047	0.007

CON, control diet group. BET, betaine diet group. EE, ether extract. TG, triglyceride. TC, total cholesterol. ^a, b^ mean in the same row with different letter superscripts signal a significant difference, ^a^ as the highest numerical value (*p* ≤ 0.05).

## Data Availability

The data presented in this study are available from the corresponding author upon request.

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
