# Peer review of "Effect of Betaine Diet on Growth Performance, Carcass Quality and Fat Deposition in Finishing Ningxiang Pigs"

_animals, 2021, doi:10.3390/ani11123408_

Round 1

Reviewer 1 Report

The manuscript describes results of trial aiming to explore the effects of dietary betaine in the growth performance, carcass quality and fat metabolism in finishing Ningxiang pigs. The trial appears to be well performed; however, a major effort needs to be done in the Introduction and Discussion section to describe the antecedents of the trial, as well as the discussion of the own results.

Line 14.& Line 55- 43.6 ±

Line 37.- Delete As we all know

Line 40.- avoid colloquial message  “Recent years has witnessed”

Introduction lacks of focus. If Ningxiang pigs show problems of performance or meat quality, introduction should give specific examples of these problems (quantitative and with references). If Betaine has been previously used in other pig breeds with the same purpose, examples should be described in the antecedents. Maybe experiment with other high fat breeds, such as Iberian pigs are relevant. A hypothesis should be declared before proposing objectives.

Table 1.- TP? AP? Change chaff by straw

Line 58. Basal diets

Line 73.- what was the criterium to choose the pig? Heaviest? or ligthest?

Line 91.- why you did not analyzed composition in LT samples??  Is not intramuscular fat interesting in Ningxiang pigs?

Line 112-113.- If there were not significant differences, the manuscript should not talk about an increase at different percentages. Results are that no significant differences were observed.

Line 133.- Total cholesterol. - avoid abbreviations if not necessary

Line 140.- LT , avoid abbreviations if not necessary

Line 158-174.- Avoid a new introduction and the justification of tested parameters. Discussion section aims to discuss your own results, either not significant or significant. The generic references to “stresses” (line 161) is confusing.

Discussion again is very confusing and needs a certain order; which it is not necessarily the same order that tested parameters. Please, focus first the fat metabolism in muscle and peripheral fat depots as a primary effect of betaine.  Try to find the way to describe the results in an easy way avoiding a list of results and comparison after each gene. Unfortunately, not IMF content in muscle is provided, don´t assume values.  Did you try to analyze IMF in Longissimus toracis or other muscles?  After discussing these effects, it is easier to understand the discussion on performance and meat quality.

Conclusion. Line 246, your results don´t show an improved growth performance.  Line 249-251 is not a conclusion.

Results Tables in general. SEM values should give one decimal more than average means

Author Response

Dear editors and reviewers:
Thank you for your letter and comment entitled "The effect of betaine diet on the growth performance, carcass quality and fat deposition of fattening Ningxiang pigs". These opinions are very valuable for the revision and improvement of our paper, very helpful, and also have important guiding significance for our research. We carefully studied the comments and made corrections. The revised part is marked in red in the manuscript. The main corrections in the paper and the responses to the reviewers’ comments are as follows: Comments and suggestions to the author

The manuscript describes results of trial aiming to explore the effects of dietary betaine in the growth performance, carcass quality and fat metabolism in finishing Ningxiang pigs. The trial appears to be well performed; however, a major effort needs to be done in the Introduction and Discussion section to describe the antecedents of the trial, as well as the discussion of the own results.

Line 14.& Line 55- 43.6 ±
A:Line 17&Line 68 A: Thanks for the reviewer’s suggestion, the wording has been revised.

Line 37.- Delete As we all know
A: Thanks to the reviewer for the suggestion, the description has been deleted.

Line 40.- avoid colloquial message  “Recent years has witnessed”
A: Thanks to the reviewer for the suggestion, the description has been deleted.

4.Introduction lacks of focus. If Ningxiang pigs show problems of performance or meat quality, introduction should give specific examples of these problems (quantitative and with references). If Betaine has been previously used in other pig breeds with the same purpose, examples should be described in the antecedents. Maybe experiment with other high fat breeds, such as Iberian pigs are relevant. A hypothesis should be declared before proposing objectives.

Line 39-57. A:Thanks for reviewer’s suggestion, the introduction of the article has been revised.

5.Table 1.- TP? AP? Change chaff by straw

Table 1.A:Thanks for reviewer’s suggestion, the description has been revised.

6.Line 58. Basal diets

Line 70-71. A:Thanks for reviewer’s suggestion, the description has been revised.

7.Line 73.- what was the criterium to choose the pig? Heaviest? or ligthest?

Line 86 . A:Thanks for reviewer’s suggestion, the description have been modified.

8.Line 91.- why you did not analyzed composition in LT samples??  Is not intramuscular fat interesting in Ningxiang pigs?

Line 106&153-160. A:Thanks for reviewer’s suggestion, the analysis of LT has been added.

9.Line 112-113.- If there were not significant differences, the manuscript should not talk about an increase at different percentages. Results are that no significant differences were observed.

Line 135-136. A:Thanks for reviewer’s suggestion, the description has been revised.

10.Line 133.- Total cholesterol. - avoid abbreviations if not necessary

Line 162-163. A:Thanks for reviewer’s suggestion, the description has been modified.

11.Line 140.- LT , avoid abbreviations if not necessary

Line 171-172. A:Thanks for reviewer’s suggestion, the description has been modified.

12.Line 264-267. Avoid a new introduction and the justification of tested parameters. Discussion section aims to discuss your own results, either not significant or significant. The generic references to “stresses” (line 161) is confusing.

Line 263-266. A:Thanks for reviewer’s suggestion, the description has been modified. 

13.Discussion again is very confusing and needs a certain order; which it is not necessarily the same order that tested parameters. Please, focus first the fat metabolism in muscle and peripheral fat depots as a primary effect of betaine.  Try to find the way to describe the results in an easy way avoiding a list of results and comparison after each gene. Unfortunately, not IMF content in muscle is provided, don´t assume values.  Did you try to analyze IMF in Longissimus toracis or other muscles?  After discussing these effects, it is easier to understand the discussion on performance and meat quality.

Line 242-249. A:Thanks for reviewer’s suggestion, the discussion have been revised and added the analysis of Longissimus toracis.

14.Conclusion. Line 246, your results don´t show an improved growth performance. 

Line 268 A: Thanks for the reviewer’s suggestion, and revised the description based on the suggestion. 

15.Line 249-251 is not a conclusion.

A: Thanks to the reviewer for the suggestion, the description has been deleted. 

16.Results Tables in general. SEM values should give one decimal more than average means

A: Thanks to the reviewer for the suggestion, the SEM value of the form has been revised. 

Reviewer 2 Report

The authors focused on the dietary supplementation of betaine in on growth performance and meat quality in finishing Ningxiang pigs. The argument is interesting since, nowadays, lean meat is largely required on the market, and strategies to improve meat quality that meets the consumer needing are required. However, the manuscript cannot be accepted in the present form. In general, many sentences are not clear. The English language should be improved. The discussion section have to be extensively revised since it is a bit misleading reporting many statements not well connected with results.

In particular, I have many suggestions to improve this manuscript as detailed below:

Simple summary

Simple summary can be expanded including more information on experimental trial and which particular parameters were influenced by the dietary supplementation.

Abstract

This section should include a brief introduction and the aim of the study.

Include a brief description of materials and methods.

Line 14: Revise the English.

Line 21: Upregulated.

Line 22: Downregulated.

Lines 22-23: All the acronyms should be provided with in extenso the first time that they appear in the text.

Line 23: Upregulated.

Introduction

In this section the hypothesis of the study has to be clearly stated.

Line 30-31: This sentence is not clear, please rephrase.

Line 31: Please clarify “worse effect”.

Line 36-37: This sentence needs a reference.

Line 38: I do not agree with the following sentence: “eccessive fat deposition will cause the imbalance of fatty acids”. The fatty acids imbalance is depending on which type of fatty acids are accumulated. It is well known that animal fats are mainly saturated. This sentence needs to be rephrased.

Line 44: Why it is necessary to report that betaine adjusting osmotic pressure and provides methyl? This sentence is not connected.

Materials and methods

Did you analyse the nutritional composition of the experimental diets?

Line 52-53: Please provide the authorization number from ethical committee.

Line 55: Revise the English.

Line 56: Were the pigs balanced per sex?

Line 57: Please, clarify what do you mean with “meal-based diet”?

Table 1: The ingredients should be listed from the most to the least concentrated. The level of Ether Extract and Ashes should be listed in nutrient levels. Each acronym should be listed in the table footnotes.

Line 66: Clarify how you calculated the nutrient levels.

Line 88: Please provide a reference for the cooking loss method.

Line 96: Please provide the list of all considered genes with their extended form followed by the acronyms.

Line 102: housekeeping gene.

Statistical analysis

Did you verified the normal distribution of collected data?

When multiple measurements are performed on the same animal or sample, a repeated measurements statistical approach should be considered.

Statistically significant differences should be defined for p 0.05.

Results

Please substitute “Result” with “Results”

Line 112-114: Since the results are not statistically significant, an increase in zootechnical performance was not observed.

Line 116: Since no superscript are listed in Table 3, the fist sentence can be deleted.

Table 4: a-b should be listed in alphabetical order. If you refer the letter “a” as the highest numerical value, this should be clarified in the table footnotes.

Discussion

This section needs to be extensively revised. In this section, a brief description of each observed result needs to be provided and successively discussed. It is important to explain why a particular result has been observed. There are many statements that do not support observed results.

In order to improve the discussion, I have some suggestions:

1 Try to explain why the observed differences in meat a*, b* and L* can be positive for the consumer and meat quality.

2 Why have you discussed about meat pH since you did not measure it.

3 Is the NAFLD important in veterinary field for animal health? Or is it more relevant for human medicine, and you are referring to the possibility to translate the use of betaine also in human nutrition?

4 What do you mean with “Compared to fat, lean meat is more popular and more expensive”?

Line 158-170: This paragraph is more an introduction than a discussion.

Line 161-162: This sentence needs to be rephrased.

Line 162: The use of term “alter” is misleading.

Line 165-166: This sentence is too trivial. Please provide more information of betaine mechanism.

Line 171: You should compare you results with previously described in literature. E.g. “our results are in line with…”.

Line 172-173: This sentence is too general.

Conclusions

The conclusions are too ambitious. Are you sure that you demonstrated that 0.2% is a reference value for betaine use in livestock?

I strongly suggest modulating your conclusions basing on what you obtained from your study.

Author Response

Dear editors and reviewers: Thank you for your letter and comment on our manuscript entitled "The effect of betaine diets on the growth performance, carcass quality and fat deposition of fattening Ningxiang pigs". These opinions are very valuable for the revision and improvement of our paper, very helpful, and also have important guiding significance for our research. We carefully studied the comments and made corrections. The revised part is marked in red in the manuscript. The main corrections in the paper and the responses to the reviewers’ comments are as follows: 

The authors focused on the dietary supplementation of betaine in on growth performance and meat quality in finishing Ningxiang pigs. The argument is interesting since, nowadays, lean meat is largely required on the market, and strategies to improve meat quality that meets the consumer needing are required. However, the manuscript cannot be accepted in the present form. In general, many sentences are not clear. The English language should be improved. The discussion section have to be extensively revised since it is a bit misleading reporting many statements not well connected with results.

In particular, I have many suggestions to improve this manuscript as detailed below:

Simple summary

Simple summary can be expanded including more information on experimental trial and which particular parameters were influenced by the dietary supplementation.

A: Lines 7-14. A: Thanks to the reviewers for their suggestions, a brief summary has been revised

Abstract

This section should include a brief introduction and the aim of the study.

Include a brief description of materials and methods.

A: Lines 15-31 A: Thanks to the reviewers for their suggestions, the abstract has been revised.

Line 14: Revise the English.
A: Line 16. A: Thanks for the reviewer's suggestion, the English has been revised.

Line 21: Upregulated.
A: Line 28. A: Thanks to the reviewer for the suggestion, the wording has been revised.

Line 22: Downregulated.
A: Line 29 A: Thanks for the reviewer’s suggestion, the wording has been revised.

Lines 22-23: All the acronyms should be provided with in extenso the first time that they appear in the text.
A: Lines 29-32. A: Thanks to the reviewers for their suggestions, all acronyms are attached when they first appear in the main text.

Line 23: Upregulated.
A:Line 31. A: Thanks for the reviewer’s suggestion, the wording has been revised.

Introduction

In this section the hypothesis of the study has to be clearly stated.
A:Lines 55-56. A: Thanks to the reviewer for the suggestion, assuming it has been stated.

Line 30-31: This sentence is not clear, please rephrase.
A:Lines 39-42. A: Thanks to the reviewer for the suggestion, the description has been rewritten.

Line 31: Please clarify “worse effect”.
A:Lines 41-42. A: Thanks to the reviewer for the suggestion, the description has been clarified.

Line 36-37: This sentence needs a reference.
A:Line 45 A: Thank you for the reviewer's suggestion. References have been added.

Line 38: I do not agree with the following sentence: “eccessive fat deposition will cause the imbalance of fatty acids”. The fatty acids imbalance is depending on which type of fatty acids are accumulated. It is well known that animal fats are mainly saturated. This sentence needs to be rephrased.
A: Line 50 A: Thanks to the reviewer for the suggestion, the description has been rewritten.

Line 44: Why it is necessary to report that betaine adjusting osmotic pressure and provides methyl? This sentence is not connected.
A: Thanks for the reviewer's suggestion, this sentence has been deleted.

Materials and methods

Did you analyse the nutritional composition of the experimental diets?

A: Thanks for reviewer’s suggestion, nutrient levels were calculated according to the standard of feed composition and nutritional value in China (30th edition in 2019). We are so sorry that we didn’t analyze the nutritional composition of the experimental diets in this study. We will put your suggestion into our future study. Thank you again.

8.Line 52-53: Please provide the authorization number from ethical committee.

Lines 66-67. A: Thanks for reviewer’s suggestion, the authorization number from ethical committee has been provided.

9.Line 55: Revise the English.

Lines 68-71. A: Thanks for reviewer’s suggestion, the English has been revised.

10.Line 56: Were the pigs balanced per sex?

Line 70. A: Thanks for reviewer’s suggestion, the related description has been added.

11.Line 57: Please, clarify what do you mean with “meal-based diet”?

Lines 70-71. A: Thanks for reviewer’s suggestion, the description have been clarified.

12.Table 1: The ingredients should be listed from the most to the least concentrated. The level of Ether Extract and Ashes should be listed in nutrient levels. Each acronym should be listed in the table footnotes.

Table 1:A: Thanks for reviewer’s suggestion, the data of table has adjusted. The level of Ashes was not measured in the study.

13.Line 66: Clarify how you calculated the nutrient levels.

Lines 79-80. A: Thanks for reviewer‘s suggestion, the description has been clarified.

14.Line 88: Please provide a reference for the cooking loss method.

Lines 104. A: Thanks for reviewer‘s suggestion, the reference has been provided.

15.Line 96: Please provide the list of all considered genes with their extended form followed by the acronyms.

Lines 119-122. A: Thanks for reviewer‘s suggestion, the list of all considered genes with their extended form has been provided.

16.Line 102: housekeeping gene

Line 122. A: Thanks for reviewer‘s suggestion, the description has been revised.

Statistical analysis

1.Did you verified the normal distribution of collected data?

When multiple measurements are performed on the same animal or sample, a repeated measurements statistical approach should be considered.

Statistically significant differences should be defined for p ≤ 0.05.

Lines 128-132. A: Thanks for reviewer‘s suggestion,the statistical analysis has been revised.

Results

1.Please substitute “Result” with “Results”

Line 133. A: Thanks for reviewer‘s suggestion, the description has been revised.

2.Line 112-114: Since the results are not statistically significant, an increase in zootechnical performance was not observed.

Lines 135-136. A: Thanks for reviewer‘s suggestion, the description has been revised.

3.Line 116: Since no superscript are listed in Table 3, the fist sentence can be deleted.

Lines 138-139 A: Thanks for reviewer‘s suggestion, the description has been deleted.

4.Table 4: a-b should be listed in alphabetical order. If you refer the letter “a” as the highest numerical value, this should be clarified in the table footnotes.

 Lines 144-145. A: Thanks for reviewer‘s suggestion, the description has been revised.

Discussion

1.This section needs to be extensively revised. In this section, a brief description of each observed result needs to be provided and successively discussed. It is important to explain why a particular result has been observed. There are many statements that do not support observed results.

 A: Thanks for reviewer‘s suggestion, the section of discussion has been revised.

In order to improve the discussion, I have some suggestions:

1 Try to explain why the observed differences in meat a*, b* and L* can be positive for the consumer and meat quality.

 Lines 226-231. A: Thanks for reviewer‘s suggestion, references have been provided to explain it.

2 Why have you discussed about meat pH since you did not measure it.

Lines 235-238. A: Thanks for reviewer‘s suggestion, the description has been revised .

3 Is the NAFLD important in veterinary field for animal health? Or is it more relevant for human medicine, and you are referring to the possibility to translate the use of betaine also in human nutrition?

A: Thanks for reviewer‘s suggestion, the discussion of NAFLD has been deleted.

4 What do you mean with “Compared to fat, lean meat is more popular and more expensive”?

A: Thanks for reviewer‘s suggestion, the description has been deleted.

5.Line 158-170: This paragraph is more an introduction than a discussion.

Lines 262-266.A: Thanks for reviewer‘s suggestion, the description has been revised.

6.Line 161-162: This sentence needs to be rephrased.

Lines 262-266.A: Thanks for reviewer‘s suggestion, the description has been rephrased.

7.Line 162: The use of term “alter” is misleading.

Line 262.A: Thanks for reviewer‘s suggestion, the description has been rephrased.

8.Line 165-166: This sentence is too trivial. Please provide more information of betaine mechanism.

A: Thanks for reviewer‘s suggestion, the description has been rephrased.

9.Line 171: You should compare you results with previously described in literature. E.g. “our results are in line with…”.

Lines 264-266.A: Thanks for reviewer‘s suggestion, references have been provided to compare our results with previously described in literature.

10.Line 172-173: This sentence is too general.

A: Thanks for reviewer‘s suggestion, the description has been rephrased.

Conclusions

1.The conclusions are too ambitious. Are you sure that you demonstrated that 0.2% is a reference value for betaine use in livestock?

I strongly suggest modulating your conclusions basing on what you obtained from your study.

Lines 268-271. A: Thanks to the reviewer for the suggestion, the description has been rewritten.

Round 2

Reviewer 1 Report

The manuscript needs of a careful revision of the English language. Please ask for a professional revision.

Introduction

Line 38.- Introduction focus unnecessarily in Energy, being the problem the metabolism and feed intake

Line 53.- did yoy speculate or hypothesized?

Line 60.- Betaine source? Company, country,…

Line 60-62.- Very diverse information in only 3 lines

Table 4.- There is no need to write different letter superscripts to sign significant differences between two treatment if the P values already inform about that. Letter are useful when there is more than 2 treatments

Discussion.

Reorganize discussion

Line 210-212 should be written, above at the start of the discussion about fat metabolism. First you bring the results that you are going to discuss about: 1.-  the likely mechanism (line 216). 2.- considering the function of each gene and previous literature reports, and 3.- linking these gene expression values to fat content in tissues found in your trial and others (see for example line 237)

Line 224. Use the verbs in the right tense. Yu et al found

Line 239. Lack the verb

Line 254.-  metabolize?

I miss the discussion of table 4 and the close link with performance results. Probably the best place is between line 218 and 219.

Author Response

请参阅附件。

Reviewer 2 Report

The authors improved the quality of this manuscript from the first submission. Hoverer, there are still some flaws that need to be corrected. In particular, the statistical analysis was not modified. Only a short sentence was added.

Find some comments and suggestions below:

In general, please add the space between terms and brackets.

Abstract: The information on methods is still missing. If results are not statistically significant, the p-value should be avoided.

Line 40: This statement requires a reference.

Line 78: The reference has to be converted into MDPI citation style.

Statistical analysis: The repeated measurements approach for zootechnical data was not included in this version of the manuscript and no answers were given to my comment.

Results: When no statistically significant results are reported the p-value should be avoided.

Discussion: This section should follow the ordering of the M&M and Results chapters. Try to reorder it according to previous sections.

Line 228: Is it considered better a lower *b24h? In this section, it was reported that meat colour was affected, but do you consider this change as positive or negative?

Author Response

请参阅附件。

Round 3

Reviewer 1 Report

Authors have made an effort to provide changes for all the comments and suggestions included in my second review. The manuscript is now acceptable for publication.

Author Response

Dear Editors and Reviewers:

Thank you for your letter and comments concerning our manuscript entitled “Effect of Betaine Diet on Growth Performance, Carcass Quality and Fat Deposition in Finishing Ningxiang Pigs” . Those comments are all valuable and very helpful for revising and improving our paper, as well as the important guiding significance to our researches. Thank you very much for taking time out of your busy schedule to review my paper, and thank you for your recognition for my paper.

Reviewer 2 Report

The authors replied to my comments and modified the manuscript according to my suggestions.

I have only some minor comments to clarify some points along the text.

Line 63: Please correct the typo (double comma)

Table 1: Try to be consistent when listing the ingredients. I suggest to use two decimal numbers for each feed ingredient.

Line 235: Try to avoid too many repetitions of "subcutaneous fat of back". Try to merge the sentence.

Table 4-5-7: In tables it is important to highlight the statistically significant different values. Asterisks or lowercase letters should be used in these cases.

Figure 1: I recommend using the same scale for y-axis in order to facilitate the reading of results.

Author Response

请参阅附件。
